# Effect of the Addition of Different Natural Waxes on the Mechanical and Rheological Behavior of PLA—A Comparative Study

**DOI:** 10.3390/polym15020305

**Published:** 2023-01-06

**Authors:** Mónica Elvira Mendoza-Duarte, Iván Alziri Estrada-Moreno, Erika Ivonne López-Martínez, Alejandro Vega-Rios

**Affiliations:** 1Centro de Investigación en Materiales Avanzados, SC, Av. Miguel de Cervantes #120, Chihuahua 31136, Mexico; 2CONACyT-CIMAV, SC, Av. Miguel de Cervantes #120, Chihuahua 31136, Mexico

**Keywords:** PLA, natural waxes, mechanical properties, rheological, thermal

## Abstract

In this study, poly(lactic acid) (PLA) blended with different natural waxes (beeswax, candelilla, carnauba, and cocoa) was investigated. Different wax amounts, 3, 5, 10, and 15 wt%, were incorporated into the PLA using a Brabender internal mixer. The blends were characterized by thermogravimetric analysis (TGA), differential scanning calorimetry (DSC), rotational rheometer (RR), dynamic mechanical analysis (DMA), and contact angle to observe the effect of the different waxes on the PLA physicochemical, rheological, mechanical behavior, and wetting properties. The complex viscosity of the blends was studied by employing a RR. The effect of the addition of the waxes on the mechanical properties of PLA was evaluated by DMA in the tension modality. A slight decrease in the thermal stability of PLA was observed with the addition of the waxes. However, in the case of the mechanical properties, the cocoa wax showed a considerable effect, especially in the elongation at break of PLA. Likewise, waxes had an essential impact on the water affinity of PLA. Specifically, with the addition of cocoa, the PLA became more hydrophilic, while the rest of the waxes increased the hydrophobic character.

## 1. Introduction

The increased demand for disposable items has caused countless ecological problems due to high pollution and the difficulty of recycling. Therefore, biodegradability is an essential factor to consider in the design of polymeric materials with only one use, including packaging, for the food industry. Polylactic acid (PLA) is a hydrophobic, aliphatic, biodegradable polyester produced by the polymerization of lactic acid molecules [1,2,3]. However, there are still some challenges in obtaining PLA-based sustainable blends as improved processability, enhanced heat resistance, and superior mechanical properties [4]. A solution to produce biodegradable materials with suitable mechanical properties is the generation of polymer mixtures. In particular, mixing PLA with naturally sourced plasticizing agents improves polymer flexibility and processing [1,5]. However, the plasticizer chosen for PLA should fulfill some requirements such as biocompatibility with PLA, stability at the processing temperatures, reducing the glass transition of the amorphous domains of the polymer, and decreasing the melting point of the crystalline phase for enhancing the molecular mobility of the PLA chains [1,2,3,5,6,7].

Additionally, the incorporation of lipids or wax in polymer films provides flexible domains within the film. The result can be a plasticizing effect, including a reduction of film strength and an increase in film flexibility [8,9]. Polymer plasticization is the disruption of polymer-polymer interactions and the replacement with plasticizer-polymer interactions increasing the free volume within the polymer structure, allowing more movement of the chains and hence increasing flexibility [10]. It is reasonable that the plasticizing agent used in the mixture with biopolymers is preferably biodegradable, for example, natural waxes. Waxes are nonpolar lipids, so they are highly hydrophobic and insoluble in water, only dissolving in typical organic solvents [5,11]. Natural ones are from plants, insects, animal skin, or mineral origin [12]. Waxes have different physicochemical properties according to their source and chemical composition [13], enhancing the final properties of wax-based composites or blends [14,15]. Beeswax (BW) is an example and is the result of a metabolic process of bees. It is moderately hard and a bit sticky and plastic [15]. It has hydrophobic characteristics and is compatible with cosmetic, pharmaceutical, and food contact applications [16,17,18]. Diyana et al. [19] blended thermoplastic cassava starch with BW in ranges from 0 to 5 wt%, reducing the tensile strength, elongation, and flexural strength while the tensile modulus was improved. Zou et al. [20] fabricated antiadhesion membranes of BW/PLA using blending electrospinning, obtaining a hydrophobic surface.

Moreover, candelilla (CLLA) wax is a vegetable wax that comes from the leaves and stems of the candelilla plant (Euphorbia antisiphilitica), which grows wild in the desert, especially in northwestern Mexico and southern Texas [15,21,22,23]. It possesses remarkable hardness, few crystallinities, and tackiness at high temperatures [13]. Few reports on polymeric blends with CLLA that study mechanical, rheological, and thermal properties exist. For instance, Janjarasskul et al. [24] prepared whey protein isolate, glycerol, and water blends in a twin-screw co-rotating extruder and added CLLA wax at 0, 5, and 7.5 g CLLA/100 g dry mix founding that CLLA has a minor effect on tensile properties of compressed sheets of the blends.

Carnauba (CBA) wax (Copernicia prunifera) originates from the Brazilian carnauba palm tree and has many applications in different products, including polishing wax, cosmetics, and dentistry. It is the hardest, highest-melting natural commercial wax [25]. CBA has excellent emulsification properties and an excellent oil binding capacity [15]. Some studies have focused on studying CBA as a compatibilizer of different blend systems [21,26,27].

Compared to conventional plasticizers based on phthalates and phenols, natural plasticizers are renewable, biodegradable, and have low toxicity without affecting their primary function of improving flexibility and facilitating polymer processing by lowering its glass transition temperature (T_g_). Although there are several PLA papers, there are few in the direction of this research in understanding the mechanical, rheological, and thermal properties of PLA–wax (BW, CLLA, CBA, and cocoa (CCA)) blend. Furthermore, it is challenging to develop completely sustainable products that do not adversely affect beings and the environment [4].

In the present study, blends of PLA with natural waxes (BW, CLLA, CBA, and CCA) were developed to modify the mechanical properties of PLA, reducing its fragility and increasing its flexibility. Likewise, studying the plasticizing effect of these waxes in a PLA matrix to develop a biodegradable blend with different properties (mechanical and wetting) to fulfill the requirements of particular applications, e.g., in the fabrication of disposable cups or as compatibilizer different polymeric blends.

## 2. Materials and Methods

### 2.1. Materials

Semicrystalline PLA 3001D was provided by NatureWorks (Plymouth, MN, USA) with a density of 1.24 g/cm^3^ and an MFR (210 °C, 2.16 kg) 22 g/10 min. The natural waxes: beeswax (BW), candelilla (CLLA), carnauba (CBA), and cocoa (CCA), were provided by Reactivos de Laboratorio (Requilab, Chihuahua, Chihuahua, Mexico). These materials were used as received.

### 2.2. Preparation of Mixtures

Prior to blending and formulating, the PLA resin was dried in an Isotemp Vacuum Oven model 281A stove for 12 h at a temperature of 70 °C. Subsequently, the PLA blends were prepared under different concentrations at 3, 5, 10, and 15 wt% of each wax (BW, CCA, CBA, and CLLA). Mixing was carried out in a Brabender internal mixer (BB) model DDRV501 (C.W. Brabender Instruments Inc., Hackensack, NJ, USA) at a temperature of 190 °C and a total mixing time of 8 min employing CAM-type blades. Additionally, the mixing programming method had a speed of 30 rpm during the first 2 min and then increased to 50 rpm for 6 min. Similarly, the neat PLA was formulated.

### 2.3. Film Preparation

The mixtures obtained were allowed to cool to room temperature, to be ground using a blade mill, Fritsch model Pulverisette. A sample (4.5 g) of pellets was taken to prepare a thin film using a hydraulic compression molding press, (Carver Inc., Wabash, IN, USA) at a temperature of 190 °C. The sample was placed between two metal plates and pressed for 3 min without applying force; afterward, 2000 kg of force was applied for 2 min. Subsequently, the material was quickly cooled to 10 °C using a cooling bath with water and left for 5 min to prevent the formation of crystals in the PLA.

### 2.4. Characterization

The thermal stability and degradation temperature of the raw materials and their blends were evaluated employing a TGA-DTA model G600-0439 (TA Instruments, New Castle, DE, USA). The analysis was carried out in the temperature range from 25 °C to 800 °C in an argon atmosphere at a flow rate of 50 mL/min. The sample weight for each analysis was 24 mg ± 0.4 mg.

About the glass transition temperatures (T_g_) and the melting temperatures (T_m_) of the formulations and their raw materials were determined using a modulated DSC equipment model 2920 (TA Instruments, New Castle, DE, USA) under inert atmosphere and a method with one cycle. The samples were heated at a temperature ramp of 10 °C/min starting from 30 °C and up to 200 °C.

The rheological properties of raw materials and the PLA–wax blends were accomplished using a rotational rheometer Physica model MCR 501 (Anton Paar, Graz, Austria). The melt viscosity of the raw materials was determined from 1 s^−1^ to 100 s^−1^ at 85 °C, using a cone-plate geometry of 40 mm in diameter, a cone angle of 1° relative to the plate, and a truncation of 48 μm. A small-amplitude oscillatory shear (SAOS) was applied for dynamic measurements, employing the parallel plate geometry, with a plate diameter of 25 mm, a gap of 1 mm, and 190 °C. Strain sweeps were scanned to define the linear viscoelastic regime. Dynamic frequency sweep tests were achieved in a frequency range of 0.01 Hz to 100 Hz in the zone of linear viscoelasticity (0.1% strain). Each measurement was repeated at least three times to minimize the experimental error.

Tensile testing was the focus of the study, evaluating the elasticity and its elongation and the effect of the addition of the waxes on the mechanical properties of PLA. The mechanical characteristics were determined from the films obtained, utilizing a mechanical dynamic analyzer (DMA) model RSA III (TA Instruments, New Castle, DE, USA). Tests were conducted on rectangular samples with a gauge length section of 12 mm × 4 mm × 0.1 mm (L × W × T). Five specimens of each formulation were analyzed. For each sample, a stress-strain test was performed in tensile mode. The applied crosshead speed was 0.01 mm/s at a temperature of 25 °C.

Contact angle (θ) analysis of films was measured in an FTA1000 contact angle tensiometer (First Ten Anstroms, Inc., Portsmouth, VA, USA). The image was captured after 60 s and analyzed with the FTA32 Software (First Ten Anstroms, Inc., Portsmouth, VA, USA). The tests were conducted at room temperature and in five different areas. Five drops were deposited in each sample using distilled water as contact liquid onto the surface. Each drop was measured, and the average and standard deviation were calculated.

## 3. Discussion and Results

### 3.1. Thermogravimetric Analysis

Thermal stability is a fundamental property when evaluating these wax-containing mixtures due to the decomposition of the aliphatic chain in correlation with temperature [28]. For this reason, the decomposition or degradation temperature is analyzed when the mixtures have a weight loss of 10%. Table 1 and Figure 1 display the temperature at 5% weight loss (T_5%_), 10% weight loss (T_10%_), and the temperature at the maximum degradation rate (T_max_). The thermograms of the PLA–wax (beeswax (BW), candelilla wax (CLLA), carnauba wax (CBA), and cocoa wax (CCA)) blends are shown in Appendix A.

An analysis of the T_5%_, compared to neat PLA, the T_5%_ was similar to the PLA–CBA and PLA–CCA blends regardless of their concentration (see Figure 1a). However, with the addition of BW and CLLA, the thermal stability of their blends diminishes. PLA–BW 15%, as an example, has a T_5%_ of 295 °C, a decrease of 32 °C. On the other hand, T_10%_ for the PLA matrix is 337 °C (see Figure 1b). With the addition of the four types of wax, the blends at T_10%_ were generally decreased. Compared with PLA, the BW showed that the magnitude of T_10%_ is correlated with the concentration in the blend, e.g., at 15 wt% BW, the diminish of T_10%_ is 23 °C. Previously, Lim et al. [29] reported a decrease in PLA degradation temperature of 2.1 °C adding 1 wt% of BW attributed to the increased mobility of PLA polymer chains.

Concerning the T_max_, Figure 1c, of neat PLA chains occurs around 361.9 °C; however, a significant change occurs with the addition of waxes. For instance, the PLA–CLLA 15 wt%, PLA–CBA 15 wt%, and PLA–BW 15 wt% blends show T_max_ values at 352.3 °C, 353.3 °C and 353.5 °C, respectively. A decrease in the T_max_ was associated with increased polymer chain mobility of the sample [1]. Furthermore, the behavior of T_max_ regarding the concentration effect of each wax on the PLA thermal stabilization is different. The T_max_ for BW tends to increment as the content of wax augments until a value of 10 wt%. After this concentration, the T_max_ decreases, suggesting that a saturation effect begins. On the contrary, the T_max_ of CLLA and CBA waxes present a decrease in the temperature as a function of wax concentration, suggesting that the saturation occurs at a concentration lower than 3 wt%. Similar behavior was observed in PLA/Epoxidized chia oil, PLA/citrate esters, and PLA/ATBC blends, where the T_max_ decreases as the plasticizer content increases, attributing it to a plasticizer saturation [2,3,4].

Notwithstanding the effect of adding waxes into the PLA, thermal stability does not induce a significant weight loss below 200 °C (<1 wt%). Since the processing temperature of this blend is 190 °C, it is therefore considered that PLA employed in this study can be mixed with any of the 4 types of wax using the conventional processing techniques and their transformation into products [30].

### 3.2. Differential Scanning Calorimetry

The glass transition temperatures (T_g_) and melting temperatures (T_m_) for the raw materials and blends from the first temperature scan were obtained using DSC (see Table 2). Regarding the melting temperature of waxes, CCA wax reported the lowest melting temperature (35.4 °C) followed by BW (62.6 °C), CLLA wax (77.3 °C), and CBA (83.4 °C), while for neat PLA was 171 °C. According to its T_m_, the choice of a plasticizer should be as close to the T_m_ of PLA to improve its processability [31]. Under this criterion, the processability in PLA–wax blends is in the following order: CBA > CLLA > BW > CCA.

Kulinski et al. [22] reported that plasticizers with a low molecular weight enhance the reduction of PLA’s T_g_, achieving an effective plasticization at relatively low plasticizer content. However, the findings obtained in our study show that only BW presents a decrease in T_g_ depending on the concentration of the blend. Specifically at 5 wt% with a T_g_ of 52.9 °C, a value approximately 15% inferior to the T_g_ shown by neat PLA (61.8 °C). This decrease may be because BW is composed of molecules of low molecular weight and, therefore, much lower than the molecular weight of PLA. Thus, these small molecules allow movements of the PLA chains at a lower temperature due to the free volume and reduced chain interactions acting as a plasticizer [23]. In relation to the rest of the waxes, where no significant decrease in PLA’s T_g_ was observed as the content of wax in the blend was increased, it could be attributed to a wax saturation effect [2,5].

Figure 2 displays a comparison of the DSC thermograms of neat PLA and PLA–waxes blends. Figure 2a shows the thermograms of the PLA–BW mixture under all concentrations. BW has a melting point close to the T_g_ of PLA. Therefore, the thermograms of PLA–BW blends present both phenomena in the same region. For PLA–CLLA and PLA–CBA blends, Figure 2b,c, the waxes have a melting temperature above the T_g_ of PLA and are observed in their respective thermograms. Finally, CCA wax has a melting point below the T_g_ of PLA (see Figure 2d). Therefore, the T_g_ of the PLA–CCA blend is appreciated acceptably more than the rest of the waxes.

### 3.3. Rheological Properties

The shear viscosity (η_0_) of raw waxes obtained using rotational rheometry at 85 °C is illustrated in Figure 3. η_0_ is directly proportional to molecular weight; therefore, the molecular weight order according to the viscosity of waxes is CBA > CLLA > CCA > BW.

The dynamic rheological analysis of polymer blends and composites is an efficient method to provide insight into their microstructures, interfacial interactions, and processing behaviors [32,33]. The complex viscosity |η*| and the storage modulus (G′) are two sensitive parameters reflecting the composite structural phase [34]. The |η*| and shear-thinning behavior of the PLA–wax composites as a function of the wax content from 3 to 15 wt% are shown in Figure 4. At low frequencies, the overall |η*|behavior of the PLA–wax blends gradually diminishes with the increasing of the wax content compared with PLA. The amount of wax directly influences the melt viscosity of PLA. The reduction in melt viscosity of the blends is ascribed to an increased free volume due to the plasticization using waxes [35]. PLA–CLLA 3 wt%, Figure 4b, and PLA–CCA 3 wt%, Figure 4d, presented higher complex viscosity at a lower frequency than PLA. On the contrary, at this content, CCA and CLLA waxes behave as reinforcing agents of the PLA matrix. However, at high frequencies, PLA–CCA 3%, and PLA–CLLA 3% blends exhibited a lower complex viscosity than PLA. In addition, at high frequencies, the PLA–BW 10% blend presented similar complex viscosity to the PLA–BW 5% blend (Figure 4a).

The viscosity behavior of the PLA–wax blends regarding the wax content at low- and high-frequency ranges is reported in Table 3 and Table 4, respectively. In the low-frequency region, PLA and their blends with 3 wt% of wax exhibit a Newtonian plateau, whereas higher frequencies present a shear-thinning behavior. In addition, the shear-thinning behavior is presented throughout the frequency range for PLA–wax blends containing from 5 to 15 wt%, representing a typical characteristic of the pseudoplastic fluid behavior. Except for PLA–CCA 5% wax and PLA–CCA 10% wax, both exhibit a Newtonian plateau at low frequencies and a shear thinning behavior in the high-frequency range [34].

Additionally, the blends prepared with CCA wax exhibited a higher complex viscosity than the analyzed blends in the low- and high-frequency range. In contrast, at the low-frequency range, the lowest viscosity was observed at 3 and 5 wt% for the blends formulated with BW. Furthermore, at 10 and 15 wt%, PLA–CBA wax had the lowest viscosity values. At the high-frequency range, the inferior viscosity was presented, at 3, 10, and 15 wt%, for the blends prepared with CBA wax. At 5 wt%, the lowest viscosity was exhibited by PLA–BW.

### 3.4. Viscoelastic Properties

The elastic modulus (G′) and loss modulus (G′′) behavior of the PLA–wax blends as a function of the wax content (from 3 to 15 wt%) are displayed in Table 5 and Table 6. The PLA–waxes blends, in general, exhibit an increase in the modulus in correlation with the frequency and typical behavior of the polymers as a consequence of the contribution of the energy necessary to produce enhanced mobility of the polymer chains [1]. A comparison of the G′ and G′′ magnitude, the behavior can be defined: as viscoelastic, solid, or liquid. When G′ > G′′, the system is dominated by viscoelastic solid-like behavior. In contrast, if G′ < G′′, has a viscoelastic liquid-like behavior [36]. As can be seen, all polymer blends exhibit a typical viscoelastic behavior (G′′ > G′) in the whole frequency range. The G′ is more sensitive than G′′ to any changes in the micro- or nanostructure of viscoelastic systems in frequency sweep tests. For this reason, G′ is analyzed in the present study. Generally, G′ is related to the elasticity of the microstructure, and the enhancement in G′ represents the enhanced elastic response of the melt under shear conditions [35].

The presence of interactions between components can alter the viscoelastic behavior of the PLA matrix. At low frequencies, the G’ of the PLA–waxes blends showed a progressively increasing function of the wax content, except for PLA–CBA blends, in which G′ is similar regardless of the wax content. Nonetheless, at high frequencies, G′ decreased with wax increased in the PLA matrix. Therefore, waxes behave as plasticizers at high and low frequencies as reinforcing agents of the PLA matrix.

The G′ behavior at a low and high-frequency range of the PLA–wax blends is depicted in Table 5 and Table 6. In addition, the PLA–CCA blends exhibited a higher elastic modulus at all concentrations than the other PLA–wax blends. The neat PLA had the lowest G′ values, indicating a less elastic nature.

### 3.5. Dynamic Mechanical Analysis (Tensile Tests)

The influence on the mechanical properties of the wax addition to the PLA matrix was evaluated. Figure 5 shows the tensile stress vs. strain curves obtained for neat PLA, PLA–BW, PLA–CLLA, PLA–CBA, and PLA–CCA blends. These results show that the most significant effect of adding wax to the PLA matrix is illustrated by the percent elongation property of the films (see Figure 5). The PLA–CCA wax blends reported the major elongation at break values of all samples (around 30.5% in PLA–CCA 3% blend) compared with neat PLA. In contrast, the addition of the other waxes did not significantly affect the elongation capacity of PLA. Therefore, CCA wax contributed to the superior mobility of the PLA polymeric chains to the other studied waxes. Some investigations exist related to the improvement of flexibility PLA. For example, Kang et al. [35] sought an environmentally sustainable plasticizer for PLA, employing cardanol, a renewable natural cashew nutshell liquid (CD); they found that blends containing more than 10 wt% of CD exhibited an improvement in flexibility attributed to a significant increment in chain mobility a hence in elongation at break. Carbonell et al. [37] reported that elongation at break for PLA-maleinized cottonseed oil blends improved to 5.8% with the only addition of 2.5 wt% of plasticizer. Janjarasskul et al. [24], in their research about tensile properties of whey protein–CLLA wax films, observed that, even when the CLLA wax level was augmented, the tensile strength decreased. At the same time, the deformation and modulus did not change significantly. They attributed this behavior to the small quantity of added CLLA (7.5%). However, as we can observe in Figure 5b, when CLLA is added in higher concentrations, it tends to lower the mechanical properties because of its elastic nature [8]. In relation to maximum stress or tensile strength at yield, for PLA, it was around 4.5 × 10^7^ Pa, and it was diminished with the addition of BW, CLLA, and CCA waxes at all concentrations. Except for the PLA–CBA blend, the tensile strength was not affected at low concentrations, specifically at 3 wt% and 5 wt%. In addition, no improvements in elongation were detected in PLA–CBA blends.

The mixtures with higher mechanical responses, such as deformation, stress, and elastic modulus, were plotted in Figure 6. In the case of maximum deformation, Figure 6a, with the addition of 3 wt% of CCA, the elongation at break, a value of around 30%, was reached, improving the plasticity of PLA. It also was found that the significant deformations achieved with the rest of the waxes were obtained when low quantities of wax were added, e.g., CBA 3%, BW 5%, and CLLA 5%. This finding could be attributed to the soft and viscous nature of CCA and BW, which offer less resistance to deformation than the high-melting waxes, such as CBA and CLLA waxes [8].

The maximum strength for different PLA–waxes blends is shown in Figure 6b. With the addition of plasticizers, the maximum stress tends to decrease due to the decrement in intermolecular forces between polymeric chains and wax. Compared to neat PLA, the PLA–CLLA 3% and PLA–CBA 3% blends presented similar values of the orders 44 MPa (4.4 × 10^7^ Pa) and 39 MPa (3.9 × 10^7^ Pa), respectively. On the contrary, the lower maximum stresses of the blends were reported by PLA–BW 3% and PLA–CCA 3%. As for the maximum elastic modulus, in Figure 6c, significant values were found for the PLA–CBA 3% and PLA–CLLA 3% blends as a consequence of their hard nature [8].

Carbonell et al. [37] reported a lower tensile strength (43.9 MPa) and elastic modulus (11.8 × 10^8^ Pa) values for the PLA/maleinized cottonseed oil (2.5 wt%) blends founded by neat PLA 46.8 MPa (4.6 × 10^7^) and 1.64 Gpa (16.8 × 10^8^ Pa), respectively.

### 3.6. Contact Angle Analysis

The wetting properties of the PLA–wax blends were investigated employing water contact angle measurements. The values for neat PLA are compared with PLA–wax blends in Figure 7. Neat PLA reports an average value of 86.3°, a slightly higher value than the one reported by Galvez et al. [1], with a value of 70.1° in an extrusion grade PLA with higher viscosity. Regarding PLA–wax blends, the contact angle decreases around 20% when 15 wt% of CCA is added. This behavior could be attributed to the hydrophilic functional groups that constitute CCA [38]. Therefore, the contact angle reduction achieved in the PLA–CCA 15% blend is indicative of an increment in the hydrophilic nature of the blend, which could favor its compatibility with other hydrophilic polymers [1].

On the other hand, with the addition of BW, CBA, and CLLA to the PLA matrix, the contact angle increased hydrophobicity close to 21% for the PLA–BW 10% and PLA–CBA 10% blends. Ferri et al. [39] reported an increment of around 13% in PLA epoxidized fatty acid esters blends, which means our blends possess higher resistance to water adsorption, making them attractive for use in the packaging industry.

## 4. Conclusions

This work aimed to modify the properties of polylactic acid (PLA) by analyzing the viability of employing different natural waxes (beeswax (BW), candelilla (CLLA), carnauba (CBA), and cocoa (CCA)) as plasticizers for obtaining a sustainable material. The thermal stability, evaluated by TGA, does not present significant changes in thermal degradation when comparing T_10%_ and T_max_ of the PLA–waxes blends and PLA; therefore, the processing conditions (<200 °C) of the PLA are not affected. The tensile results indicated that adding CCA at 3 wt% improves the deformation capacity of PLA, increasing 416%. However, the maximum stress is decreased (44%) at the same CCA concentration. Furthermore, it is important to note that the elastic modulus of the PLA–CCA (3 to 10 wt%) blends is similar to PLA, with the advantage of experimenting with deformations superior from 25 to 35%. Hence, the blends exhibit high flexibility and mechanical resistance to PLA, which is desirable for their application to manufacture single-use plastic products, e.g., disposable cups and plates.

About with maximum stress, the mixing PLA with CLLA (3 wt%) displayed an identical value to PLA. Additionally, the processability was improved according to complex viscosity results. On the other hand, the elastic modulus can be overcome by adding any wax, except with CBA.

Clear differences in contact angle were observed when evaluating PLA–wax blends using water as the solvent. For example, the PLA–BW, PLA–CLLA, and PLA–CBA blends showed a more hydrophobic nature than PLA. On the contrary, the PLA– CCA blend recorded a considerably decrease in the contact angle than the neat PLA, suggesting more hydrophilicity of the blends. The contact angle modification was independent of the concentration of the waxes incorporated into the PLA.

## Figures and Tables

**Figure 1 polymers-15-00305-f001:**
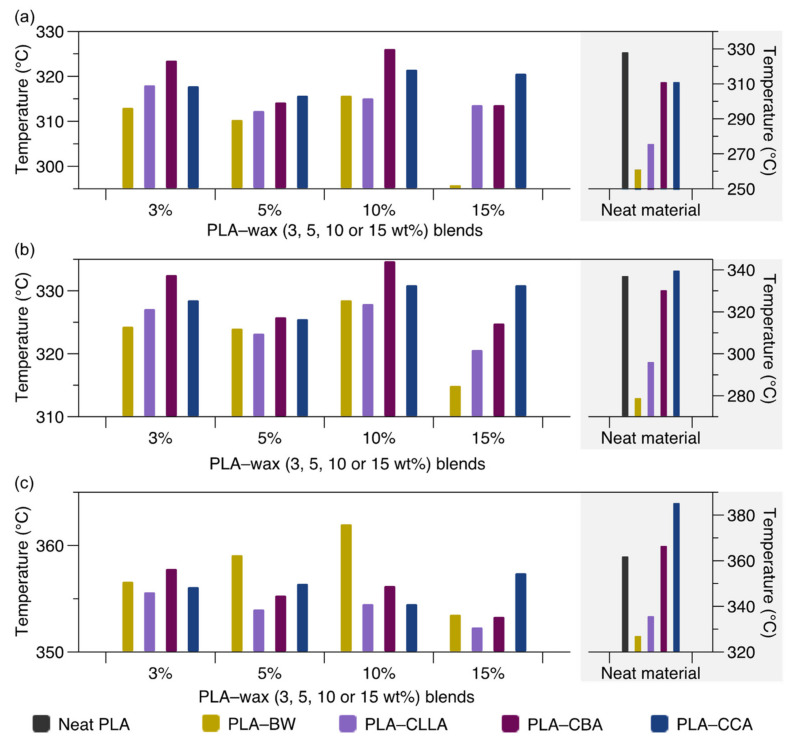
(**a**) Degradation temperature at 5% weight loss. (**b**) Degradation temperature at 10% weight loss temperature, and (**c**) maximum degradation temperature of neat polylactic acid (PLA), wax (beeswax (BW), candelilla (CLLA), carnauba (CBA), and cocoa (CCA)) and PLA–wax (BW, CLLA, CBA, and CCA)) blends and their respective neat materials.

**Figure 2 polymers-15-00305-f002:**
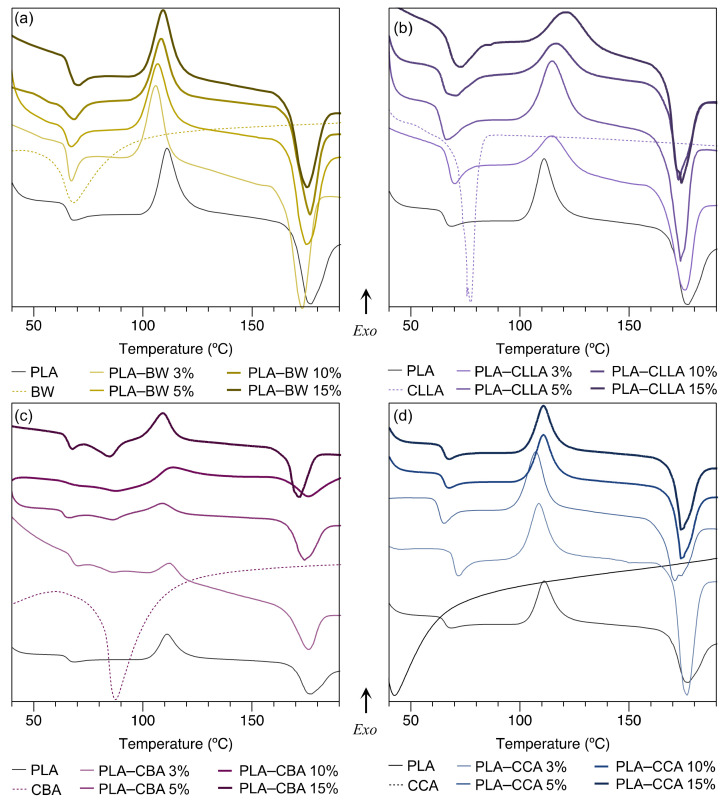
DSC thermograms of PLA–wax blends compared with PLA and its wax. (**a**) PLA–BW blend, (**b**) PLA–CLLA blend, (**c**) PLA–CBA blend, and (**d**) PLA–CCA blend.

**Figure 3 polymers-15-00305-f003:**
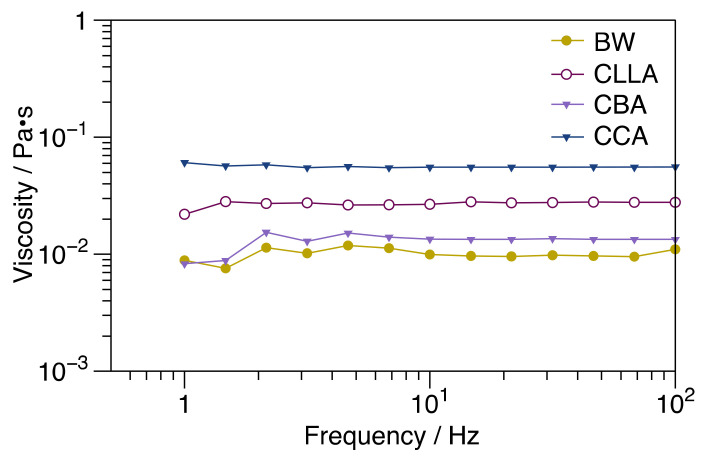
Shear viscosity raw waxes at a temperature of 85 °C.

**Figure 4 polymers-15-00305-f004:**
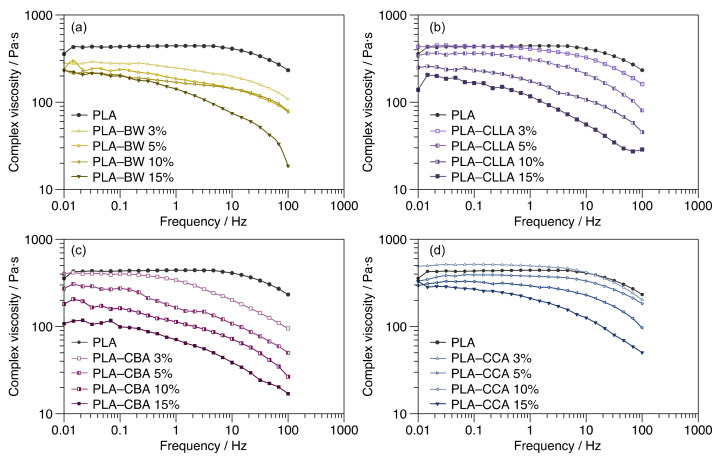
Complex viscosity of PLA–waxes (BW, CLLA, CBA, and CCA) blends at 3, 5, 10, and 15 wt% compared to PLA. (**a**) BW, (**b**) CLLA, (**c**) CBA, and (**d**) CCA.

**Figure 5 polymers-15-00305-f005:**
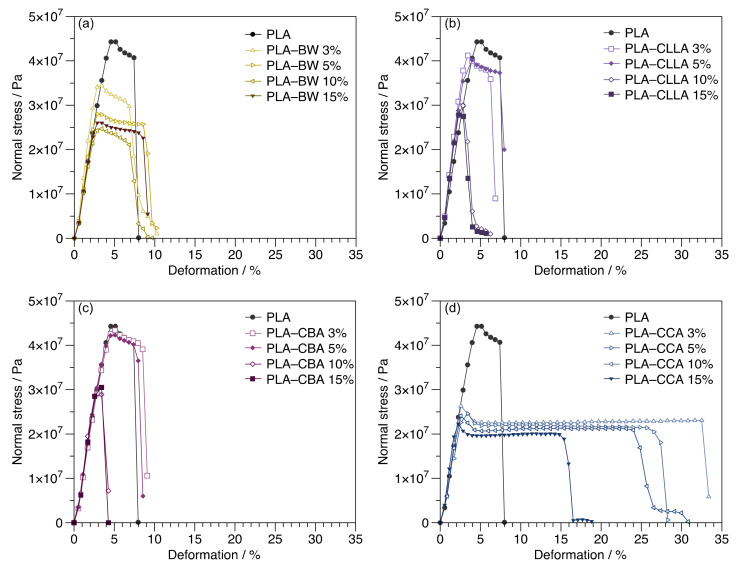
Stress–strain curves of PLA blends containing four different waxes at 3, 5, 10, and 15 wt%. (**a**) BW, (**b**) CLLA, (**c**) CBA, and (**d**) CCA.

**Figure 6 polymers-15-00305-f006:**
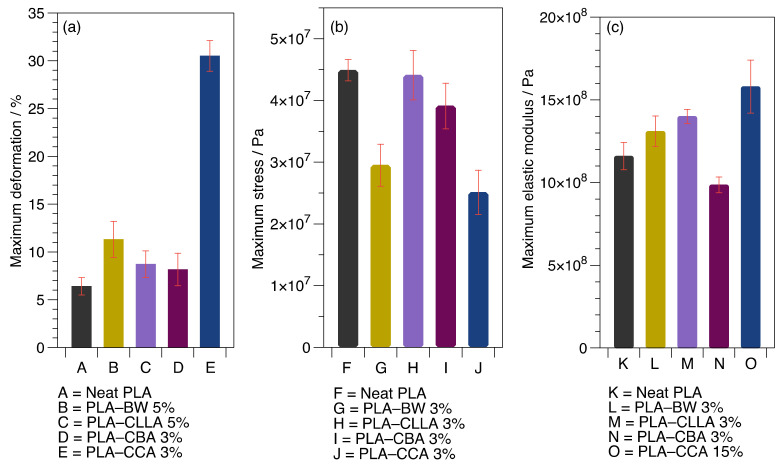
Maximum values were obtained in each set of blends. (**a**) Maximum deformation, (**b**) maximum stress, and (**c**) maximum elastic modulus.

**Figure 7 polymers-15-00305-f007:**
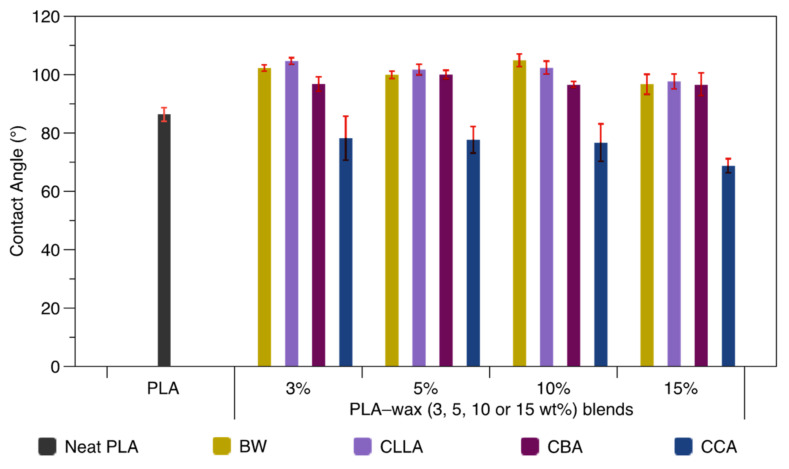
Contact angle values for neat PLA and PLA–wax blends at different concentrations.

**Table 1 polymers-15-00305-t001:** Data were obtained from the TGA thermograms of the PLA–beeswax (PLA–BW), PLA–candelilla (PLA–CLLA), PLA–carnauba (PLA–CBA), and PLA–cocoa (PLA–CCA) blends.

	Neat PLA			BW			CLLA			CBA			CCA		
	T_5%_	T_10%_	T_max_	T_5%_	T_10%_	T_max_	T_5%_	T_10%_	T_max_	T_5%_	T_10%_	T_max_	T_5%_	T_10%_	T_max_
Neat material	328.1	337.1	361.9	261.0	278.9	327.0	275.6	296.1	335.7	311.0	330.3	366.5	311.0	339.7	385.3
PLA–Wax %															
3	–	–	–	313.0	324.3	356.6	318.0	327.1	355.6	323.5	332.5	357.8	317.8	328.5	356.1
5	–	–	–	310.3	324.0	359.1	312.3	323.2	354.0	314.2	325.8	355.3	315.7	325.5	356.4
10	–	–	–	315.7	328.5	362.0	315.1	327.9	354.5	326.1	334.7	356.2	321.5	330.9	354.5
15	–	–	–	295.8	314.9	353.5	305.6	320.6	352.3	313.6	324.8	353.3	320.6	330.9	357.4

**Table 2 polymers-15-00305-t002:** DSC data from 1st scan of neat PLA; BW; CLLA; CBA; CCA; PLA–BW (3, 5, 10, and 15 wt%); PLA–CLLA (3, 5, 10, and 15 wt%); PLA–CBA (3, 5, 10, and 15 wt%); and PLA–CCA (3, 5, 10, and 15 wt%).

Sample	T_g_ ^1^ (°C)	T_cc_ ^2^	ΔH_cc_ ^3^	T_m_ ^4^ (°C)	ΔH_m_ ^5^	ΔH_m_ − ΔH_cc_
BW	--	--	--	68.3	102.3	--
CLLA	--	--	--	77.4	117.4	--
CBA	--	--	--	87.5	161.1	--
CCA	--	--	--	42.5	62.1	--
neat PLA	65.6	111.1	20.3	176.5	24.8	4.5
BW						
PLA–BW 3%	65.4	106.0	21.4	173.0	31.5	10.1
PLA–BW 5%	64.9	106.8	23.0	175.0	32.6	9.6
PLA–BW 10%	64.7	108.4	18.9	176.5	28.2	9.3
PLA–BW 15%	65.6	109.2	22.3	175.0	28.3	6.0
CLLA						
PLA–CLLA 3%	67.7	115.3	17.1	175.3	31.1	14.0
PLA–CLLA 5%	64.6	114.9	25.2	173.6	32.6	7.4
PLA–CLLA 10%	64.7	116.9	14.9	172.7	27.4	12.5
PLA–CLLA 15%	67.5	122.0	20.4	174.0	29.7	9.3
CBA						
PLA–CBA 3%	67.0	112.7	11.4	175.5	33.8	22.5
PLA–CBA 5%	63.8	109.4	8.3	174.1	32.8	24.6
PLA–CBA 10%	68.8	112.9	17.8	175.9	31.6	13.8
PLA–CBA 15%	67.5	109.2	21.2	172.1	31.2	10.0
CCA						
PLA–CCA 3%	69.4	108.7	28.0	176.4	47.9	19.9
PLA–CCA 5%	63.0	107.2	29.3	170.9	31.5	2.2
PLA–CCA 10%	65.1	110.7	22.3	173.7	27.4	5.1
PLA–CCA 15%	69.4	111.3	25.8	173.2	25.9	0.1

^1^ T_g_ glass transition temperatures. ^2^ T_cc_ Temperature cold crystallization. ^3^ ΔH_c_ Enthalpy cold crystallization. ^4^ T_m_ melting temperature. ^5^ ΔH_m_ melting enthalpy.

**Table 3 polymers-15-00305-t003:** Complex viscosity (|η*|) of PLA–wax biocomposites in the low-frequency range.

|η*| (Pa∙s)	3 wt%LF *	5 wt%LF *	10 wt%LF *	15 wt%LF *
Highest	CCA	Neat PLA	Neat PLA	Neat PLA
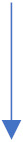	CLLA	CCA	CCA	CCA
Neat PLA	CBA	CLLA	BW
CBA	CLLA	BW	CLLA
Lowest	BW	BW	CBA	CBA

* LF = Low-frequency range.

**Table 4 polymers-15-00305-t004:** Complex viscosity (|η*|) of PLA–wax blends in the high-frequency range.

|η*| (Pa∙s)	3 wt%HF *	5 wt%HF *	10 wt%HF *	15 wt%HF *
Highest	Neat PLA	Neat PLA	Neat PLA	Neat PLA
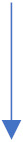	CCA	CCA	CCA	CCA
CLLA	CBA	BW	BW
BW	CLLA	CLLA	CLLA
Lowest	CBA	BW	CBA	CBA

* HF = High-frequency range.

**Table 5 polymers-15-00305-t005:** Elastic modulus (G′) of PLA–wax blends in the low-frequency range.

G′ (Pa)	3 wt%LF *	5 wt%LF *	10 wt%LF *	15 wt%LF *
Highest	BW	similar modulus	CCA	CCA
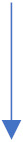	CCA	CBA	BW
CBA	CLLA	CLLA
CLLA	BW	CBA
Lowest	Neat PLA	Neat PLA	Neat PLA

* LF = Low-frequency region.

**Table 6 polymers-15-00305-t006:** Elastic modulus (G′) of PLA–wax blends in the high-frequency range.

G′ (Pa)	3 wt%HF *	5 wt%HF *	10 wt%HF *	15 wt%HF *
Highest	Neat PLA	Neat PLA	Neat PLA	Neat PLA
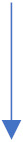	CCA	CCA	CCA	CCA
CLLA	CLLA	BW	BW
CBA	BW	CLLA	CBA
Lowest	BW	CBA	CBA	CLLA

* HF = High-frequency region.

## Data Availability

The data presented in this study are available on request from the corresponding author.

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
