# Peer review of "Effect of the Addition of Different Natural Waxes on the Mechanical and Rheological Behavior of PLA—A Comparative Study"

_polymers, 2023, doi:10.3390/polym15020305_

Round 1

Reviewer 1 Report

Overall, the authors clearly addressed the significance of the work of the effect of natural waxes on PLA thermal, mechanical, and rheological behavior. There are some typos in the manuscript. For example, in the abstract “the cacao”. Line 65, “whey” . The grammar of the writing needs to be improved.

Line 262: missing G’’ after loss modulus.

Reviewer 2 Report

Although the article deals with quite interesting issues related to the modification of polylactide with natural compounds, a major revision is necessary before accepting the text for publication. In general, the text needs to be reviewed from a linguistic side. Below are some questions/comments to which the authors should refer.

1. Why was the temperature of 10% weight loss adopted as a measure of thermal resistance? Usually, for this purpose, a temperature of 5% weight loss is assumed.

2. The authors should add a table with the results of TG research, which will greatly facilitate the analysis of the text.

3. Lines 165 - 167. Please explain the given statement more fully in the text, rather than just quoting it.

4. Fig. 1. Authors should change the scale of the y-axis so that it does not start at zero.

DSC curves for individual or selected materials are missing from the text. Without the curves, it is impossible to fully analyze the test results.

5. Line 190 – 194. With regard to the explanations presented, the question arises why, then, no changes were observed with reduced wax contents?

6. Table 1. Please group the tables according to increasing concentrations of each type of wax. In addition, you can add a drawing showing the effect of each type of wax. Such a presentation of research results would be much clearer.

7. If possible, the roughness of the obtained test samples could be determined. As it is known, this parameter strongly affects the results of contact angles. It could turn out that the obtained changes in contact angles are more related to the roughness than to the chemical changes of the surface layer.

Round 2

Reviewer 2 Report

The authors addressed all comments contained in the original review. After the changes made, I am in favor of accepting the article for publication in its present form.